# Prediction of Friction Power via Machine Learning of Acoustic Emissions from a Ring-on-Disc Rotary Tribometer

Christopher Strablegg * , Florian Summer , Philipp Renhart and Florian Grün

Montanuniversität of Leoben, Chair of Mechanical Engineering, Franz Josef-Straße 18, 8700 Leoben, Austria
* Correspondence: christopher.strablegg@unileoben.ac.at; Tel.: +43-3842-402-1454

**Abstract:** Acoustic emissions from tribological contacts have become an interesting field of science in recent years. This study focuses on predicting the friction power of a given system (lubricated ring-on-disc geometry), independently of the used sliding material and lubricant, from the acoustic emissions emitted from the system. The low-frequency (1 Hz), continuously measured RMS value of the acoustic data is combined with short-duration and high-frequency (850 kHz) signal data in form of the power spectra and hit rate with three prominence levels. The classification system then predicts the friction power of the test system continuously over the whole test time. Prediction is achieved by four different machine learning methods (tree-type, support vector machine, K-nearest-neighbor, neural network) trained with data from 54 ring-on-disc tests with high variation in material and oil combinations. The method allows for the quantifiable and step-free prediction of absolute values of friction power with accuracy of 97.6% on unseen data, with a weighted K-nearest-neighbor classifier, at any point in time during an experiment. The system reacts well to rapid changes in friction conditions due to changes in load and temperature. The study shows the high information degree of acoustic emissions, concerning the actual friction mechanisms occurring and the quantitative, and not only qualitative, information that one can gain about a tribological system by analyzing them.

**Keywords:** acoustic emission; friction prediction; machine learning

## 1. Introduction

The application of different acoustic emission (AE) methods in industry and science—for example, in material and default testing or the live condition monitoring of machinery—is already well established. It is often used in a wide variety of mechanical systems [1], in addition to other parameters, such as friction, temperature and load, or to completely substitute them because their in-situ measurement would require unjustifiable effort [2,3].

Acoustic signals from tribological processes in particular can provide various insights into different friction and wear mechanisms and events [4,5] that other easy-to-measure parameters cannot sufficiently achieve. Ensuring good signal quality and the useful acquisition of data [6] is necessary to gain these additional data. The selection of fitting postprocessing methods [7] to resolve the intended effects and phenomena plays also a significant role. Some of the established methods are as follows:

- the root-mean-square (RMS) value, which gives clues as to the overall running and surface condition [8,9];
- a variety of spectral methods, such as Fourier transform (FT), short-time FT (STFT) or wavelet transform (WT), providing information about friction mechanisms [4];
- or event-based methods such as hit rate (HR) or spectral kurtosis (SK) [7,10].

An ongoing challenge in this field of study is to make quantitative statements based on the acoustic data. Qualitative changes in tribological behavior—for example, the general lubrication (hydrodynamic friction, oil starvation, etc.) state [11], wear state (mild or severe wear) [4] or remaining lifetime [12]—can be separated in a given system with various tools

already. Nevertheless, finding quantitative connections between the acoustic emission signal and wear, friction or other relevant properties—at best, system-independent—remains for further investigation, which has been done partly in this study.

In recent years, the development of sufficient machine learning techniques opened a new path for evaluating test data of various sources (temperature, friction, load, contact potential, etc.) [13–16] from tribological processes. The technique opens the door to a better understanding of the underlying effects in tribology, even surpassing well-established simulation methods, such as predicting the film thickness of elasto-hydrodynamic lubricated contacts [17]. The implementation of the acoustic data seems to be especially useful, because of the high-frequency data and dense information content compared to classical parameters. Already, many different machine-learning based classifiers and neural networks have been tested with the aforementioned raw or postprocessed data [18,19], predicting or classifying, for example, different running states (running-in, steady-state, severe wear, etc.) [11,18], various states of wear [19,20] or the remaining lifetime of the system [21,22].

In this work, the task is to predict the friction power of a ring-on-disc (RoD) test system with acoustic data in combination with a machine learning-based method. Comparable work has been performed already in predicting COF [23,24] or modelling elasto-hydrodynamic lubrication friction by machine learning [25], but always with a limited scope in the variety of materials, lubricants or test conditions. The challenge in this work was to achieve a fine differentiation between the possible prediction results over a wide range of sliding materials, oils and test conditions. The novelty is to combine spectral and event-based data from non-continuous high-speed AE measurements, feeding the classifier, with low-speed but continuously sampled AE data in the form of the RMS value, to obtain a step-free and continuous prediction for the friction power.

The classifier predicts the ratio of AE RMS/friction power, which is afterward combined with the AE RMS value to calculate the friction power. With a high number of predictable classes (21 categories) in combination with the continuously available RMS value, the classifier is usable for a wide range of cases for the given ring-on-disc geometry. Furthermore, it shows again that the acoustic data from tribological processes seem to contain a lot of in-depth information about the tribological processes on the surface of the stressed components.

## 2. Materials and Methods

### 2.1. Experimental Set-Up

As mentioned, a ring-on-disc (RoD) set-up (see Figure 1) was used to generate the data for the classifier. In essence, the set-up simulates lubricated plain rotating contacts, mostly in the mixed lubrication regime, depending on the specific shape of both disc and ring samples.

A rotating disc slides against a fixed ring in the axial direction. Varying normal pressure can be applied from beneath, and the temperatures and rotational speeds can be adjusted. The temperature of the specimen and oil, normal load, friction force and coefficient-of-friction (COF), contact potential between ring and disc, displacement/wear of the specimen, as well as acoustic emission are monitored constantly in every test run.

The disc specimen is composed of different base materials or specific bearing materials (special polymers and alloys), while the ring is composed of ordinary steel. The base materials can be, for example, AlSi-based (e.g., found in cylinder walls) or bronze, a typical journal-bearing liner material. Additionally, the disc specimen's sliding tread is quartered by 4 notches for better oil entrainment at the contact area (see Figure 2). The overall impact of a hydrodynamic friction regime remains low; the system therefore runs mostly in a mixed lubrication regime, with boundary friction at the running-in phase and low velocity tests.

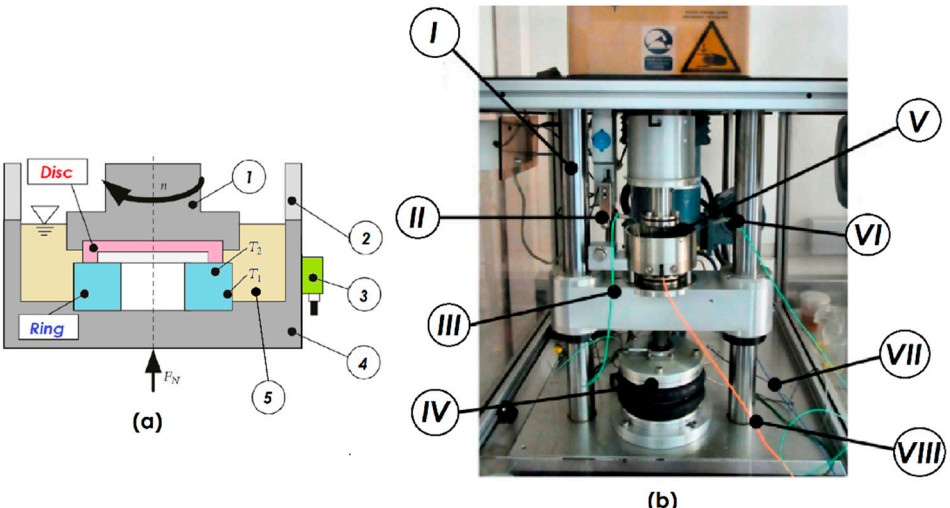

**Figure 1.** RoD test setup. (**a**) Schematic sectional image of rotational test cell: (1) rotating shaft, (2) splash guard, (3) AE sensor, (4) heated oil bath, (5) oil. (**b**) Picture of the test cell in the tribometer: (I) linear guiding struts, (II) friction force sensor, (III) specimen temperature sensor (left green cable), (IV) air bellow for force application, (V) heated oil bath, (VI) oil bath temperature sensor (right green cable), (VII) contact potential sensor (blue cable), (VIII) wear sensor (orange cable).

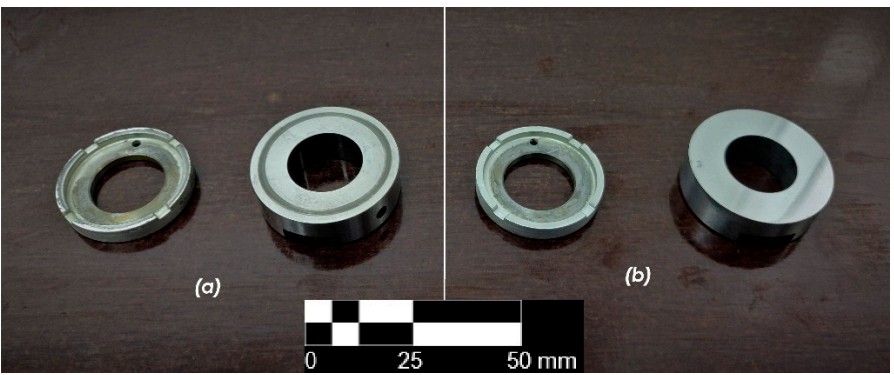

**Figure 2.** RoD specimens, steel ring (diameter d = 35 mm) and steel disc (d = 30 mm) with 4 notches with a metal-based coating; (**a**) samples after finished test run; (**b**) pristine samples before test run.

### 2.2. Acoustic Emission Measurments

As described above, a single acoustic sensor was applied at the side of the oil bath. One layer of Kapton® tape was applied between the sensor and oil bath to isolate the sensor electrically and prevent ground loops in the signal. The sensor itself was a Kistler 8152C with an effective frequency range between 50 and 900 kHz. The AE signal then passed into a Kistler 5125C Piezotron coupler, where it was transformed into a voltage signal, before filtering and amplifying it. The band-pass filter applied used a 50–1000 kHz range and the amplification factor was 1000. Then, the RMS was formed with a window size of 1.2 ms out of the filtered and amplified signal.

Both the filtered raw signal and the formed RMS signal were then transferred to an A/D high-speed capture card of type NI USB-X 6341 (see Figure 3). Sampling frequency was set to 850 kHz, reducing the usable maximum frequency for the data to 425 kHz. The relatively low frequency range compared to the possible 900 kHz of the sensor did not reduce the effective amount of information that we gained from the signals. According to previous studies [6,10], the main frequency band for AE originating from tribological phenomena for our geometry and measuring principle was 50–250 kHz.

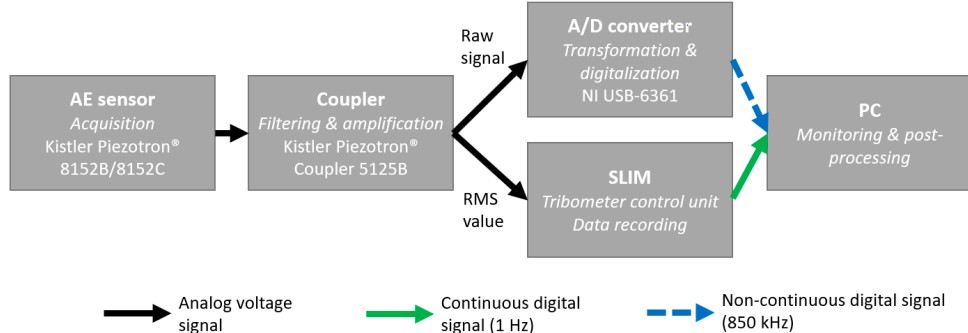

**Figure 3.** AE signal chain from sensor to coupling device and the digitalization process.

The RMS signal was digitalized with 1 Hz for the full test time, while the filtered raw signal was sampled with 850 kHz, resulting in the aforementioned 425 kHz effective signal range. The high-frequency data of the raw signal were not recorded continuously over the whole time of the test run; small, few-second-long recordings were made at certain steps of a test. Thus, the denser information of the high frequency was only available every few minutes of a test, while the RMS showed changes in the test run continuously.

### 2.3. Material and Oil Combinations

For this study, 7 different sliding materials or tribological coatings of 9 batches (production batch of samples) were used, as well as a variety of 17 oils (for simplicity, named oils 1–17) with different viscosities and additives in various combinations with the materials. This resulted in acoustic and tribological data from 54 test runs. The material and oil combinations were as follows.

- Polymer-coated samples: polymer-coated bearing material, capable of resisting high loads with minimal wear. Coating on Cu base material (oils 1–3).
- Tin–copper (SnCu) samples in three batches: Cu base material coated with a high-resistance Sn matrix with Cu inclusions (oils 1 and 4–14).
- Bronze samples: base Cu material, resulting in high wear rates and relatively poor friction behavior (oils 12–15).
- Aluminum–tin with 6% tin (AlSn6): (oil 16).
- Aluminum–tin with 20% tin (AlSn20): (oil 16).
- Pure aluminum (Al) samples: (oils 16–17).
- Aluminum–silicon (AlSi) samples: (oil 16).

The opposing steel ring was polished in a certain direction; surface roughness ranged from 0.12 to 0.15 μm Ra (arithmetic average of profile height) and 0.013 to 0.016 μm Rz (maximum peak to valley height of the profile), measured perpendicular to the direction of polishing.

This variation in the expected behavior of the bearing materials, in combination with the different oils, gave a good data basis for friction prediction. Furthermore, the friction regime and behavior should change significantly with the multi-layered material over the course of a single test, when underlying layers are exposed.

### 2.4. Test Strategies

Two different test designs concerning load levels, rotational speeds and oil bath temperatures were used. Most of the tests were conducted in a shock-load design; the rest were conducted with a constant-load test with one step (see Figure 4).

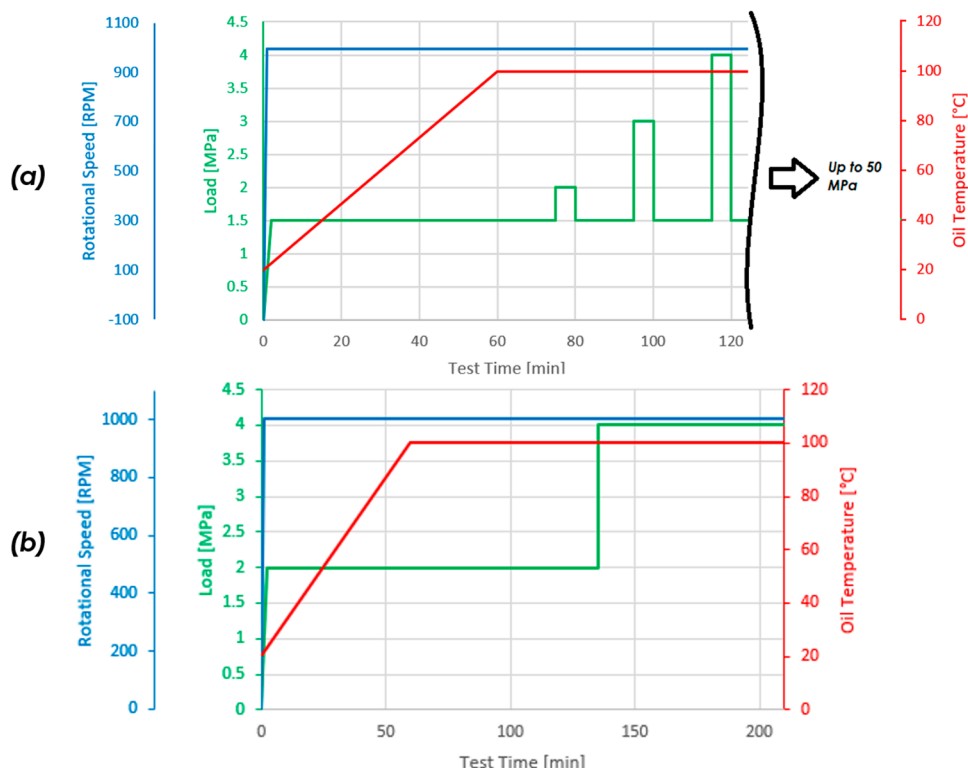

**Figure 4.** Test strategies for the RoD experiments. (**a**) The first 120 min shock-load test with the first 3 load steps. Test was continued after the visualized cut with similar load increased (1 MPa increments) in the periodic load steps up to a maximum of 50 MPa; the test finished after the maximum load step. (**b**) Complete visualization of constant-load test, which lasted around 210 min, with a maximum load of 4 MPa; afterwards, the test finished.

For the shock-load tests, the first 120 min, with the three shock loads at 2/3/4 MPa, are depicted in Figure 4, while the whole test lasted until a 50 MPa maximum load, with load increments of 1 MPa for the shock loads, as in the first 120 min. Temperature was increased to 100 °C in 60 min; a base load of 1.5 MPa and a speed of 1000 RPM were set from the beginning. The test stopped after the final load step or when it reached certain levels of COF (max. 0.15) or temperature (180 °C at specimen). This test strategy was designed to investigate the resistance of the sample against sharp load increases. Thus, we purposely induced an environment of high wear and possible seizure.

The constant-load test was designed with one step. The first load step lasted from the beginning until 135 min with 2 MPa; afterwards, the load was increased to 4 MPa. Temperature was increased to 100 °C in 60 min; speed was set at 1000 RPM from the start. Again, the test stopped when we reached the set stop time at around 210 min or when we reached certain levels of COF (max. 0.15) or temperature (180 °C at the specimen). The constant-load test was designed specifically to investigate low but constant wear.

The high-speed AE measurements were timed at the shock-load test at the end of every high load level and at the end of every low load level. Every signal had a duration of a few seconds. At the full constant-load test, the high-speed measurements were timed every 15 min (with the same signal length as above) and at the step increasing load at 135 min.

*2.5. Data Acquisition and Preperation for Classifier*

The classifier should choose the ratio of AE RMS to friction power depending on high-speed AE data. The high-speed AE data were measured as described above, each signal with a length of a few seconds and a few million data points. These signals were

then separated into 100-ms-long parts. These 100 ms samples were then processed in two different ways:

- The power spectrum was created using FT, every sample ranging in frequency from 50 to 425 kHz with 512 data points per spectrum;
- The hit rate for every 100 ms piece was calculated for 3 different levels of prominence (0.05 V, 0.25 V, 0.5 V) of the single AE peaks in the signals, resulting in 3 values; each value represents the hits per second over a certain minimum threshold.

The spectral data were then normalized via Z-score normalization, using feature scaling, while the hit rate was obtained by instance scaling.

For each of these 100 ms pieces, the actual AE RMS to friction power ratio was categorized into one of 21 categories. The thresholds of these categories were chosen manually to give a good representation of the actual values. One additional category was set as category 0, which corresponds to non-useable values, i.e., out-of-boundary or non-existent values. The category boundaries are listed in Table 1.

**Table 1.** Boundaries of categories used for classification, values for the AE RMS to friction power signal.

| Category | Requirement for AE RMS/Friction Power [V/W] |
| --- | --- |
| 0 | $<10^{-4}$, or NaN |
| 1 | $>10^{-4}$ |
| 2 | $>10^{-3}$ |
| 3 | $>10^{-2}$ |
| 4 | $>2 \times 10^{-2}$ |
| 5 | $>5 \times 10^{-2}$ |
| 6 | $>0.1$ |
| 7 | $>0.2$ |
| 8 | $>0.3$ |
| 9 | $>0.4$ |
| 10 | $>0.5$ |
| 11 | $>0.6$ |
| 12 | $>0.7$ |
| 13 | $>0.8$ |
| 14 | $>0.9$ |
| 15 | $>1$ |
| 16 | $>1.5$ |
| 17 | $>2$ |
| 18 | $>2.5$ |
| 19 | $>3$ |
| 20 | $>4.5$ |

The choice of taking the RMS/friction power value for categorization was made because it represents the change in friction mechanism better than the simple RMS value. The same friction events could occur more frequently in the test, which increased the RMS as well as the friction power, but the RMS/friction power value remained the same, indicating an unchanged mixture of friction mechanisms. This fits well with the goal of predicting the mixture of these friction mechanisms with the spectral and hit data.

In the end, the dataset had 86,000 samples, each with FFT (with 512 features), hit rate (with 3 features) and a category, which were all normalized and connected to a specific category. The following graphic (Figure 5) represents the whole workflow for data preparation.

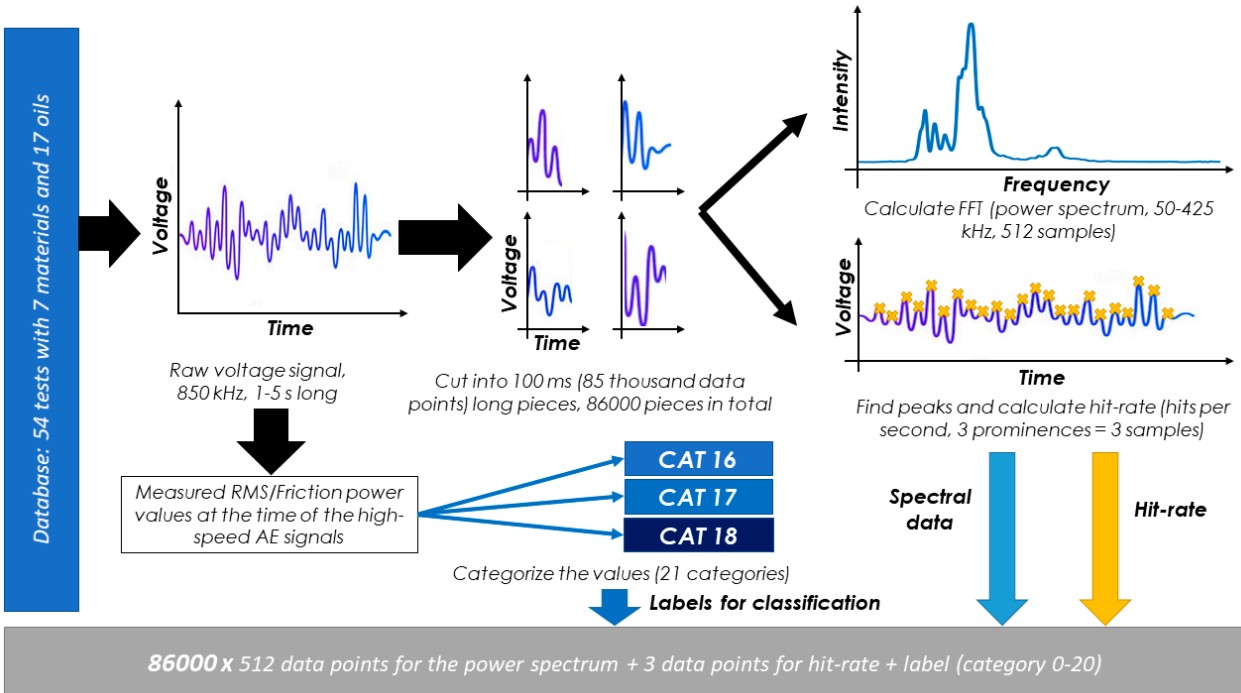

**Figure 5.** Workflow chart showing preparation of learning data for the classifier, from raw signal to usable lines of data, with labeling for the learning process of the classifiers.

Three different data sets were prepared, one with the spectral and hit rate data, one with only the spectral data and the last with only the hit rate. This was done to differentiate the impact for the accuracy of the classifier between the two postprocessing methods (FFT, hit rate).

### 2.6. Classifiers and Machine Learning Tools

Four different methods for classifying the given data set were used. These were a tree-style (TS) classifier, a support vector machine (SVM), a k-nearest-neighbor (KNN) algorithm-based system and a bi-layered neural network (BLNN). Their specific settings are listed below:

- The tree-style classifier had a maximum number of splits of 100, with Gini's diversity index as a split criterion, with surrogate decision splits disabled.
- The SVM had a cubic kernel function with an automatic kernel scale (value is optimized automatically by the program), a box constraint level of 1.0, concerning the penalty factor of the SVM, and a One-vs.-One multiclass learning method.
- The weighted KNN had 10 neighbors with the Euclidian distance metric and squared inverse distance weight and no further hyperparameter optimization.
- The neural network consisted of 2 fully connected layers. The first layer had 50 neurons, while the second had 25. The activation function was a rectified linear unit (ReLU) and zero regularization strength. Iteration limit was 1000 cycles.

### 3. Results

*3.1. Exemplary Tests and Parameter Comparison*

Figure 6 shows selected test graphs of RoD tests with the aforementioned test strategies (see Section 2.4).

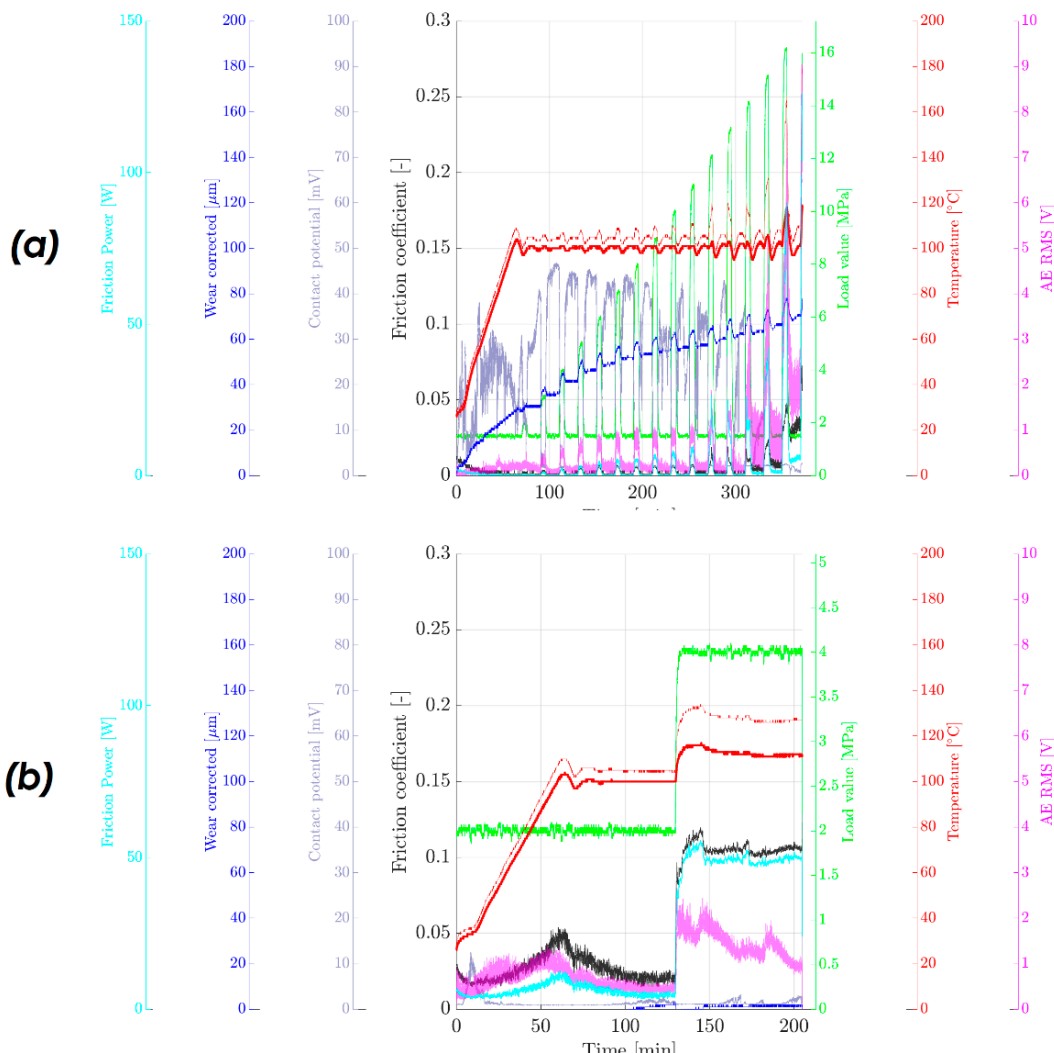

**Figure 6.** (**a**) Test graph of a shock-load test with a Cu bearing material. Note: Test failed at 17 MPa of load due to high friction values. (**b**) Complete constant-load test of an AlSi material. Test stopped at expected time.

Data from both tests were used to generate the training data for the classifiers. When looking at the friction power (cyan) and the AE RMS value (magenta) for both tests, one can clearly see the connection between the two values. Of course, also the COF (black) follows a similar trend. The measured normal load (green) shows the intendent steps already shown in Figure 4. The temperatures (dotted red = specimen temperature, solid red = oil bath temperature) react as expected to the load increases in both tests; the temperature becomes higher, and the specimen temperature sees a stronger increase due to the friction heat. The contact potential (grey) measures the electric potential between the ring and disc; 50 mV is the maximum value and means no electrical contact, while zero means direct unshielded contact between friction partners. The load spikes in the test in Figure 6a are associated with the collapse of the contact potential. Here, the buildup of tribological layers from oil additives is reduced, which exposes the surfaces to more direct contact. In-situ wear (dark blue) is also recorded and measures the relative separation of lower and upper specimen adapters. It visualizes the wear of materials during the test, being very visible in Figure 6a with the quite soft sliding material, compared to the very hard sliding material in Figure 6b.

The acoustic signal rises and falls mostly in conjunction with the friction power. All friction-related effects that contribute to the friction power generated in the system also contribute to high-frequency vibration (AE), so the behavior seems logical. Nevertheless,

the factor between the AE RMS and friction power levels is not constant over the course of the tests. This is best depicted by comparing the graphs of both values in Figure 6b after the load increase of the system at 120 min. The AE RMS increases with the friction power, but not at the same level. This observation becomes much clearer when looking at the AE RMS to friction power levels directly in the next graphic (Figure 7a,b).

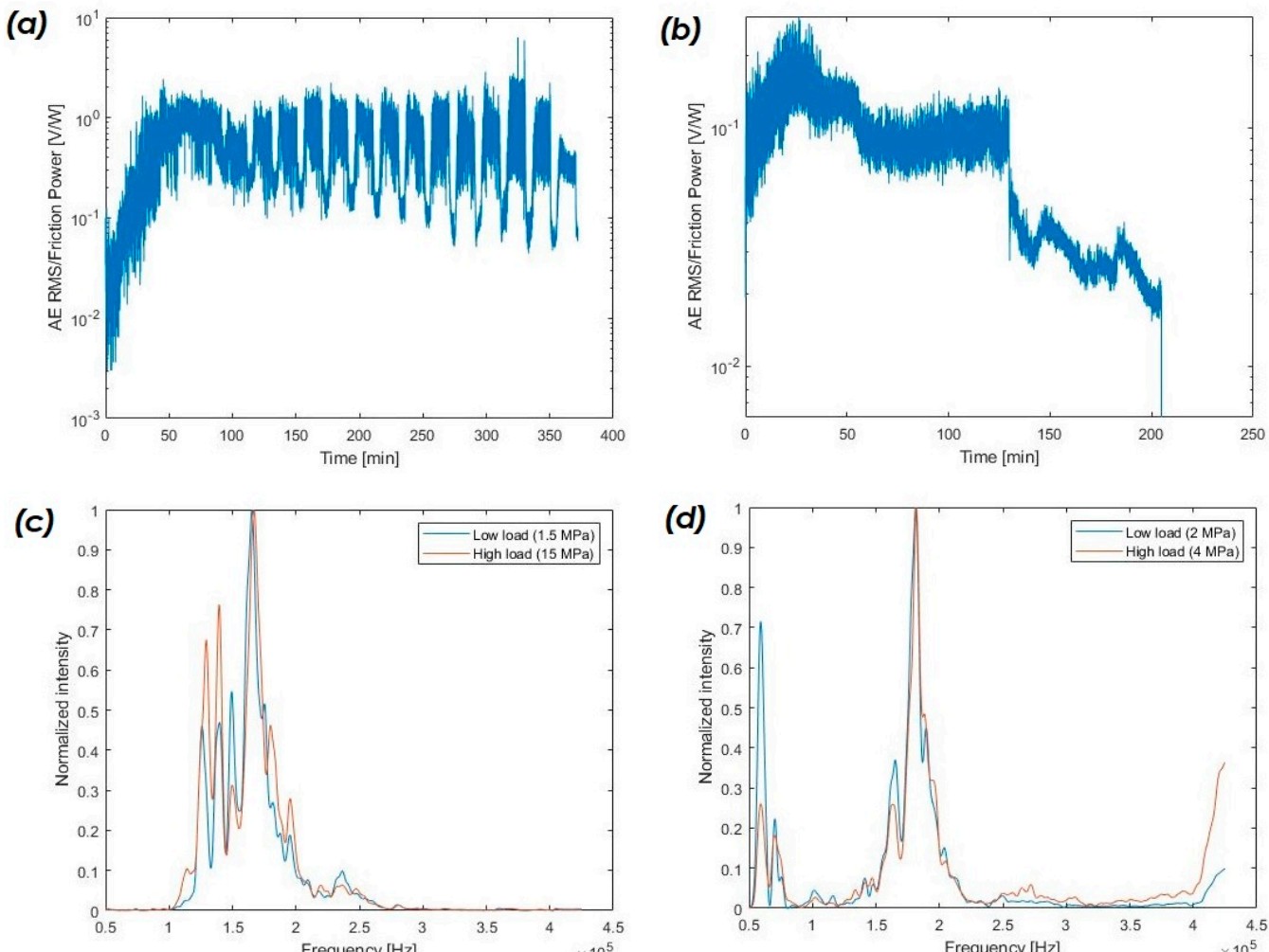

**Figure 7.** AE RMS/friction power value over the whole test time for (**a**) the shock-load test in Figure 6a and (**b**) constant-load test in Figure 6b. Intensity-normalized power spectrum of (**c**) the shock-load test in Figure 6b and (**d**) constant-load test in Figure 6a, both at different load levels.

Interestingly, the increase in load levels in both test environments led to lower RMS/friction power values. For example, in Figure 7b, at 135 min test time, the load is increased (visible in Figure 6b) and the AE RMS/friction power value decreases. Likewise, in the shock-load test of Figure 7a, the short regions with a higher load feature lower AE RMS/friction power values. Thus, in theory, the tribological processes have changed in such a way that they emit less intense AE.

This change could also be seen in the spectral composition of the signal (see Figure 7c,d). The overall spectrum composition does not change notably in both examples in the case of a load increase. However, a slight change in the weighting of the different frequency regions is visible, in addition to a strong variation in absolute intensity, which is not visible in Figure 7c,d directly. If we assume that every tribological mechanism emits AE in a certain frequency range, almost all mechanisms seem to be preserved after the load increase. What changes is the weighting in the mixture of these tribological mechanisms. Another

assumption is that every mechanism has a certain impact on friction and therefore friction power.

The problem with the spectral data is that the information in the time–space domain is completely lost, and therefore the hit rate was introduced to add additional data for decision making. The hit rate (see Table 2) increases with a load increase. The density of events rises, and the severity of events increases. In both cases, the hit rate is higher with a higher load for each of the three prominence levels. Moreover, as expected, the number of high-prominence hits rises, while, in some cases, the number of lower-prominence hits falls by comparison (see Table 2, 0.05 V hit rate in the shock-load test). Nevertheless, comparing the two tests against each other, one can note higher hit rates in the constant-load test, even with lower load levels. This shows the limitation of the hit rate method, because it cannot differentiate well between different material and oil parings under varying test procedures. The method is not capable of clearly separating different friction mechanisms compared to the spectral method, but is very useful in comparing the severity of friction.

**Table 2.** Comparison of hit rates from a shock-load test and constant-load test with different materials and lubricants at different load levels for three prominence requirements (0.05 V, 0.25 V, 0.5 V).

| | Shock-Load Test | | | | Constant-Load Test | | |
|---|---|---|---|---|---|---|---|
| Load [MPa] | Hit Rate 0.05 V | Hit Rate 0.25 V | Hit Rate 0.5 V | Load [MPa] | Hit Rate 0.05 V | Hit Rate 0.25 V | Hit Rate 0.5 V |
| 1.5 | $1.8498 \times 10^5$ | $1.6521 \times 10^5$ | $1.4539 \times 10^5$ | 2 | $2.0670 \times 10^5$ | $1.8199 \times 10^5$ | $1.5737 \times 10^5$ |
| 15 | $1.7777 \times 10^5$ | $1.7516 \times 10^5$ | $1.7228 \times 10^5$ | 4 | $2.4397 \times 10^5$ | $2.3475 \times 10^5$ | $2.2377 \times 10^5$ |

*3.2. Training of Classifiers*

As already mentioned, four types of classifiers were used (TS, SVM, KNN and BLNN). These classifiers were trained with three data sets. These were the hit rate alone, the spectral data alone and the combined data. This was selected in order to resolve the impact on the accuracy of the classifiers of each of the parameters (power spectrum and hit rate) separately.

Thus, three separate data sets, each with 86,000 instances and 512 features (power spectrum), 3 features (hit rates) or 515 features (power spectrum + hit rate), were created. The randomly mixed instances were than halved, with 43,000 for learning and 43,000 for testing. For each of the three sets, the randomization was the same. A five-fold cross-validation procedure was implemented on all classifiers to protect against overfitting. The resulting accuracy of the classifiers with the training and test data is listed below (see Table 3).

The hit rate alone seems to contain insufficient information to generate good predictions, independent of the chosen classifier. The spectral data are a much better parameter to make decisions; the prediction accuracy with the power spectrum alone is, with the exception of the TS, consistently over 90%. The addition of the hit rate as a feature boosts the accuracy from almost zero (92.9% vs. 93.2% with the BLNN) to approximately 2.4% in the case of the KNN (95.2% vs. 97.6%). Based on these data, the KNN was chosen for further investigation.

With the help of the confusion matrix in Figure 8 and the true-positive rate (TPR) and false-negative rate (FNR) in Table 4, it is noticeable that especially the differentiation between classes 2–9 is slightly problematic. These classes correspond to low RMS/friction power values, i.e., regions with high friction and low AE intensity levels. The high uncertainty of class 0 is caused by the NaN values that it contains for the friction power. Nevertheless, most of the time, incorrect classification occurs in neighboring classes, so the resulting values for friction power should be close to the true values.

**Table 3.** Accuracy of the 4 classifiers with training and testing data sets with hit rate or power spectrum and both combined.

| Tree-Type (TS) Classifier | | | |
|---|---|---|---|
| Data Set | Hit Rate (3 Features) | Power Spectrum (512 Features) | Hit Rate + Power Spectrum (515 Features) |
| train data | 54.3% | 59.7% | 61.1% |
| test data | 54.4% | 59.3% | 61.2% |
| Support Vector Machine (SVM) Classifier | | | |
| Data Set | Hit Rate (3 Features) | Power Spectrum (512 Features) | Hit Rate + Power Spectrum (515 Features) |
| train data | 8.7% | 95.2% | 96.2% |
| test data | 11.6% | 95.1% | 96.8% |
| K-Nearest-Neighbor (KNN) Classifier | | | |
| Data Set | Hit Rate (3 features) | Power Spectrum (512 Features) | Hit Rate + Power Spectrum (515 Features) |
| train data | 69.4% | 95.6% | 97.3% |
| test data | 69.4% | 95.2% | 97.6% |
| Bi-Layered Neural Network (BLNN) | | | |
| Data Set | Hit Rate (3 Features) | Power Spectrum (512 Features) | Hit Rate + Power Spectrum (515 Features) |
| train data | 57.4% | 92.7% | 92.7% |
| test data | 56.3% | 92.9% | 93.2% |

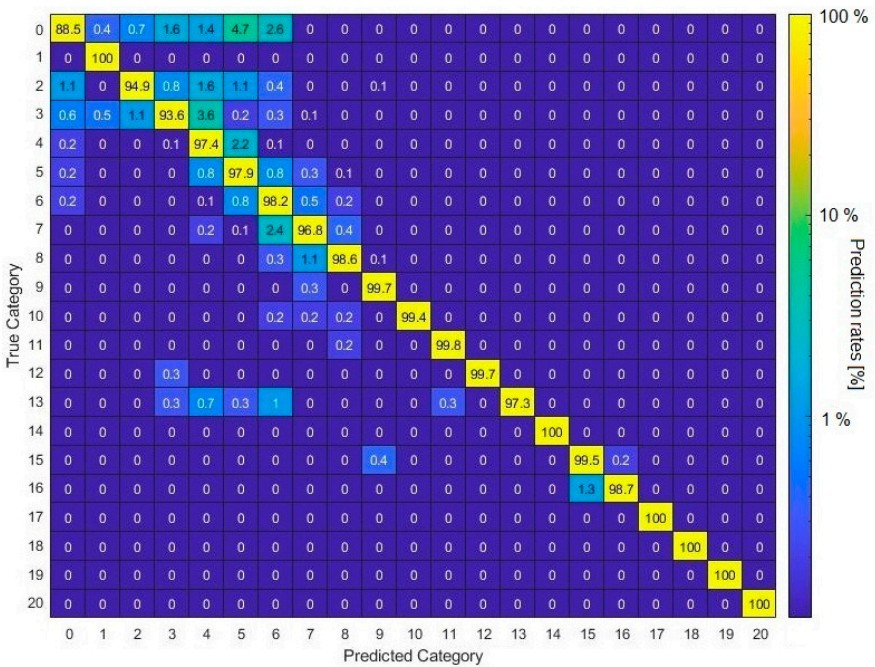

**Figure 8.** Confusion matrix with prediction rates for true and predicted categories (categories of Table 1) for the KNN classifier, for the test data using the combined hit rate and power spectrum data.

**Table 4.** True-positive rates (TPR) and false-negative rates (FNR) for the KNN classifier, for the test data using the combined hit rate and power spectrum data for all 21 categories.

| Category | TPR [%] | FNR [%] |
|---|---|---|
| 0 | 88.5 | 11.5 |
| 1 | 100.0 | 0.0 |
| 2 | 94.9 | 5.1 |
| 3 | 93.6 | 6.4 |
| 4 | 97.4 | 2.6 |
| 5 | 97.9 | 2.1 |
| 6 | 98.2 | 1.8 |
| 7 | 96.8 | 3.2 |
| 8 | 98.6 | 1.4 |
| 9 | 99.7 | 0.3 |
| 10 | 99.4 | 0.6 |
| 11 | 99.8 | 0.2 |
| 12 | 99.7 | 0.3 |
| 13 | 97.3 | 2.7 |
| 14 | 100.0 | 0.0 |
| 15 | 99.5 | 0.5 |
| 16 | 98.7 | 1.3 |
| 17 | 100.0 | 0.0 |
| 18 | 100.0 | 0.0 |
| 19 | 100.0 | 0.0 |
| 20 | 100.0 | 0.0 |

*3.3. Validation of Model*

The best prediction model (weighted KNN using hit rate and power spectrum) in the case of prediction accuracy was chosen to be implemented for the classification of two different, unseen, full-length long test runs. The chosen tests were of the shock-load type with an SnCu and bronze material (see Figure 9 for SnCu and Figure 10 for bronze). Both tests had very different levels of the AE RMS/friction power value (compare Figure 9c to Figure 10c) to show that the classifier was functional over all classes. The tests were performed on another tribometer (another motor, another structural design), but with the same adapters for sensors and samples, in the same positions relative to the tested specimens. This was specifically chosen to determine how the model behaves with new data and to rule out the possibility that the classifier generates the predictions from AE originating not from the tribological contact.

To make the predictions, a similar procedure as described in Section 2.5 was implemented. The only difference was that the few-second-long signal was not split into 100 ms parts, but used as a whole. Thus, the power spectrum and the three hit rates were calculated for each of the few-second-long fragments over the course of the test. These were then fed into the classifier. Every predicted class was then retransformed into the AE RMS/friction power value by taking the lower-class border. Because the classified value only was evaluated for a few seconds of the test, it was assumed that the AE RMS/friction value stayed the same until the next high-speed AE measurement. By multiplying it with the continuously available AE RMS value, the friction power itself was calculated (see Figures 9d and 10d).

As visible in Figures 9d and 10d, the friction power values can be predicted alone from acoustic data to a certain level of accuracy. This shows that the information of the AE must originate from the tribological contact; otherwise, changing the testing machine would have led to much worse results of the classifier. Moreover, it demonstrates that critical information about the occurring friction mechanisms, which determine the friction power, are represented in some form in the AE data and especially in the spectral data. This enables one to make quantifiable statements about the system from the AE data and not only qualitative ones.

The predicted value for the SnCu test from Figure 9a is good for lower load steps and near the points in time of the only few-second-long high-speed AE measurements. This

can be seen quite clearly in the unloaded steps, were there is some notable variation in the predicted values. Moreover, for lower load steps, the prediction seems to be better than for higher ones, firstly overestimating the friction power and later underestimating it.

Overall, the second test with the bronze material in Figure 10a shows also an accurate prediction for friction power values, but lacks the capacity to detect finer changes in friction power, as already seen in the test with the SnCu material. Moreover, one can see the limits of the method in the spikes in friction power in the load steps in Figure 10d at 140 and 150 min of test time, which were not predicted by the classifier.

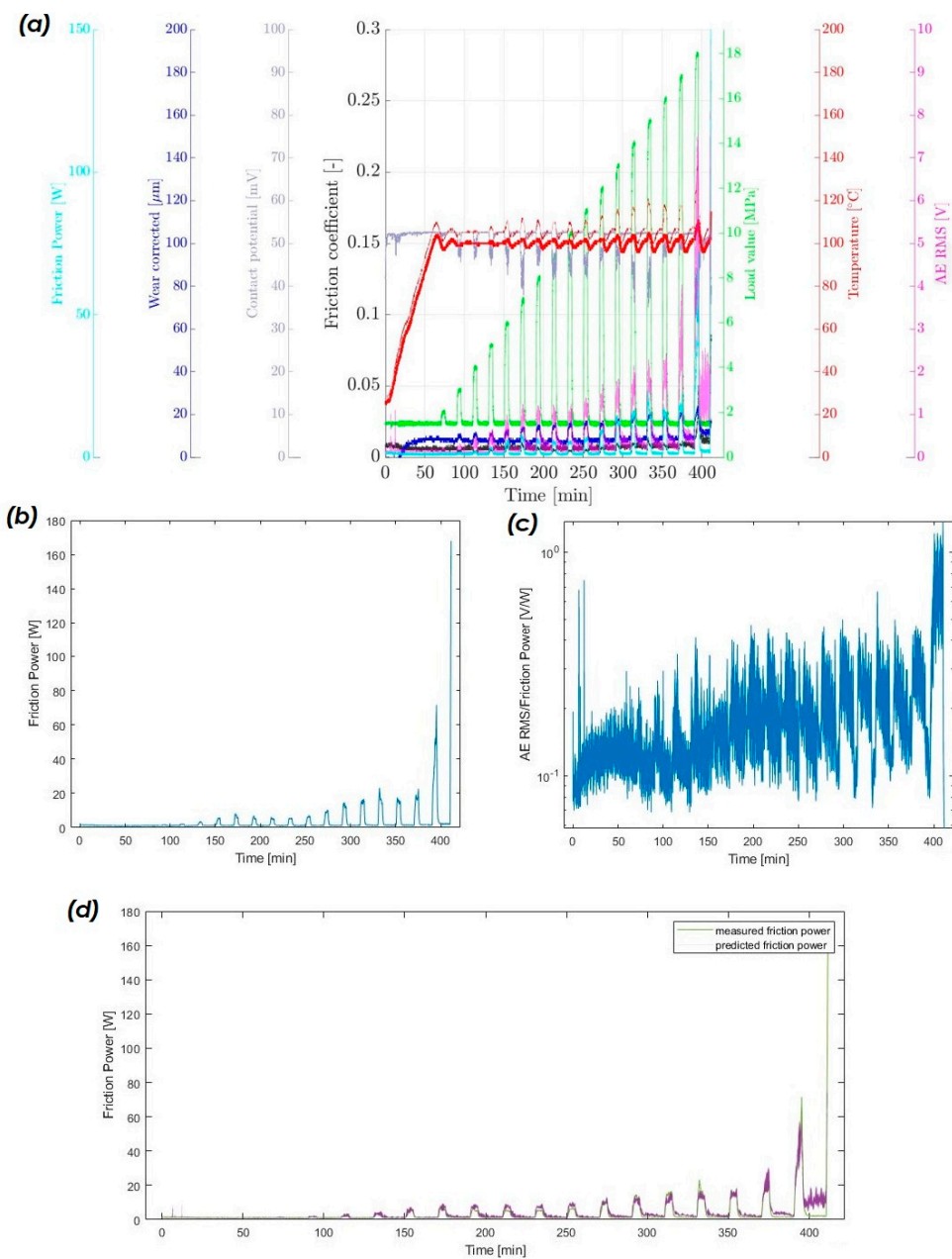

**Figure 9.** (**a**) Test overview of shock-load RoD test with SnCu material, with unseen data for the classifier; (**b**) measured friction power of the same test; (**c**) AE RMS/friction power value of the same test; (**d**) comparison of measured and predicted friction power of the test.

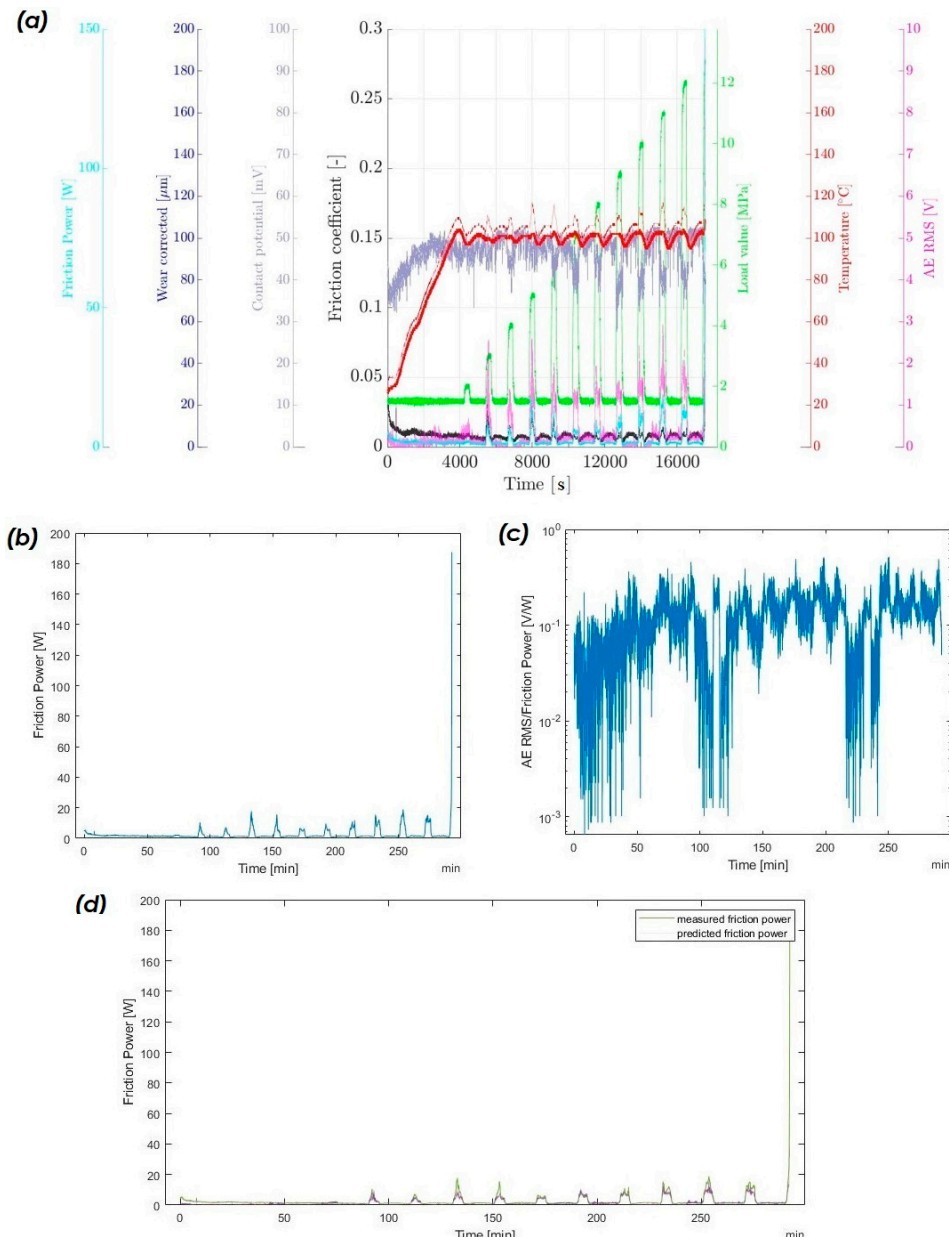

**Figure 10.** (**a**) Test overview of shock-load RoD test with bronze material, with unseen data for the classifier; (**b**) measured friction power of shock-load of the same test; (**c**) AE RMS/friction power value of the same test; (**d**) comparison of measured and predicted friction power of the test.

## 4. Discussion

In the analysis of the results of this work, some interesting insights can be gained into the topic of AE in tribology.

First, the spectral data (in the form of the power spectrum) seem to contain sufficient information to generate good predictions for AE RMS/friction power values. The frequency range of up to 425 kHz does not limit the visualization of the system. As shown in Figure 9a, most of the intensity of the signals in our tests falls within the range of 50–250 kHz, which coincides with the findings of Hase et al. [12], where they found most of the intensity of the friction-related signals below a threshold of 0.5 MHz. Moreover, in the works of Renhart et al. [10] and König et al. [11], the range of frequencies from sliding friction is concentrated even below 350 kHz. Thus, for the results for three out of the four classifiers without using the hit rate, only with the spectral data, scoring over 90% appears reasonable. No major data seem to be missing, concerning the frequency range, to make good predictions. The

misclassified entities for the three classifiers are close to the true classes. The hit rate can be a useful addition to boost accuracy (0.3–2.4% with test data), but lacks the same information density as the spectral data, at least in the form used in this work. Classifiers using only the hit rate achieve below 70% accuracy. The KNN with 69.4% accuracy worked best, while the TS and BLNN scored 54.4 and 56.3%. The SVM reacted most poorly to the limited data of the hit rate, with only 11.6% accuracy.

Essentially, the result is further confirmation that in the acoustic signal, there is more information about the friction mechanism occurring than initially expected. The spectral data and the hit rate give information about the composition of the friction mechanisms, determining the AE signal intensity released by every unit of friction power. By scaling this value with the absolute AE intensity in the form of the RMS value, one obtains the friction power. Thus, otherwise difficult-to-measure information about the tribological system can be gained with data from solely a single acoustic sensor. We can compare this to the two force sensors (one for friction torque and one for load) combined with the actual sliding speed data of the specimens needed to calculate the friction power conventionally.

Moreover, the machine learning system helps one to separate all the different frequencies occurring in the spectrum and analyze their impacts on specific tribological values, such as the friction power. Otherwise, this process would be very work-intensive or almost impossible with the given data set, because no specific tests were done to promote one friction effect over the other. The tests used for this work consisted of a mixture of friction mechanisms at all times. From previous works [6,7,10], it is known that for the used test geometries and sensors, the AE from sliding friction ranges mostly between 70 and 250 kHz, with boundary friction accruing around 90 and 140 kHz and hydromantic effects at higher frequencies of 100 kHz and180–230 kHz, as well as particle effects that could be found at around 200 kHz.

The system that was introduced in this work could be improved in several ways in the future. An FFT from the high-speed AE signal could be recorded continuously over the course of the test, so that one does not have to assume that the general friction regime is not changing between the high-speed measurements, which is currently one of the limitations of the system. The high-speed AE recordings are triggered at the beginning of the shock loads, after reaching the intended load level. The system assumes that the composition of the friction mechanisms remains constant until the next high-speed AE measurement at the low-load step afterwards. If there are any changes in the friction mechanisms during this gap in time, the system cannot react to them. This is visible in Figure 10d in the shock loads, for example, from 130 to 140 min test time. Here, the predicted value is only close to the true value at the beginning of the shock load; later, the true value exceeds the predicted value noticeably.

Another limitation is the high steps between categories 15 and 20, which are mostly prevalent in low-load steps. This has also a negative impact on the accuracy, as the true AE RMS/friction power value fluctuates near these category borders. This can be noticed in Figure 9d around 400 min test time, where the predicted value is higher than the true one. In Figure 9c, one can see that the AE RMS/friction power value in the low-load step fluctuates around 1 V/W, placing it directly between categories 14 and 15.

Additionally, further improvements could be made to the hit rate parameter, by using more features for the hit rate with a higher spread of the prominence levels of the single hits, for example. In the presented example in Table 2, for the shock-load tests, higher prominence thresholds for the peaks would be beneficial to better separate high-load and low-load scenarios. The difference for the 1.5 and 15 MPa load cases for the 0.5 V prominence level is only ~20% for a high change in frictional intensity.

## 5. Conclusions

In summary, 54 tests with 2 test strategies, 17 different oils and 9 batches of tribological materials (polymer, SnCu, bronze, AlSn6, AlSn20, Al, AlSi), were performed to generate the necessary data for this study. Different methods of classifiers, in the form

of a TS/SVM/KNN or BLNN, were used. In this work, the weighted KNN obtained the best results, predicting the friction power of a tribological system with accuracy of 97.6%. The novelty of this work lies in the step-free and continuously, over the whole test time, available prediction of the friction power only from the AE data, because of the high number of categories and the scaling with the continuously available AE RMS.

The spectral data with 512 features impact the classification results of the classifiers to a higher degree than the hit rates with only three features. A rough estimation of the true class with only the hit rate is possible in the form of the KNN classifier has almost 70% accuracy.

In two examples with previously unseen data for the classifiers, from another testing system (another motor, electrical system and structural elements), it was proven that the classification system does not fit to other systems' inputs, such as AE from the motor or electrical system. Over the selection of sliding materials, the classifier works well, but it would be interesting to test in the future how the system reacts to data from experiments with other tribological materials.

The findings of this study give rise to the further investigation of the acoustic signals of tribological origin—in particular, to further differentiate between the mechanisms of friction and wear causing adhesive effects, particle friction, impact phenomena or corrosion-related effects, to name only a few. This work focuses on the friction side of systems, but shifting the main focus to in-situ wear monitoring would be interesting in the future. Moreover, we show that AE in tribology can be very useful to obtain quantifiable measurements of the system. The experience gained with the machine learning system will be useful to further improve the support for the aforementioned goals in the future, hopefully implementing a live monitoring system with AE in an industrial environment.

**Author Contributions:** Conceptualization, C.S.; methodology, C.S and P.R.; software, C.S.; validation, C.S. and F.S.; formal analysis, C.S.; investigation, C.S.; resources, C.S.; data curation, C.S.; writing—original draft preparation, C.S.; writing—review and editing, C.S., P.R. and F.S.; visualization, C.S.; supervision, F.S. and F.G.; project administration, F.S. and F.G.; funding acquisition, F.G. All authors have read and agreed to the published version of the manuscript.

**Funding:** This research received no external funding.

**Data Availability Statement:** Not applicable.

**Conflicts of Interest:** The authors declare no conflict of interest.

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
