# Peer review of "Prediction of Friction Power via Machine Learning of Acoustic Emissions from a Ring-on-Disc Rotary Tribometer"

_lubricants, doi:10.3390/lubricants11020037_

Round 1

Reviewer 1 Report

Dear Authors! I should admit very relevant set of the problem and sound research design. I found few typos (see file attached). 

My main concern is the lack physical interpretation - it is obvious that COF and wear rate as well as friction power are related with a number of physical parameters mutually resulting in friction mechanisms. You touched this issue but did not give any clue how AE frequencies are interrelated with these particular mechanisms. 

I think these issues deserve to be discussed in Conclusions, for example.

Author Response

Dear Reviewer,

thank you for your valuable input. Beneath I want to go into detail what was changed in the work to address your comments:

  •  I found few typos (see file attached).
    The typos that were marked in the attached file were fixed.
  • My main concern is the lack physical interpretation - it is obvious that COF and wear rate as well as friction power are related with a number of physical parameters mutually resulting in friction mechanisms. You touched this issue but did not give any clue how AE frequencies are interrelated with these particular mechanisms.
    The discussion was elongated, especially in line 475-484 to go into detail of the seen frequencies and the connected physical effects. The Literature which described the connection more in-depth was cited [1-4,10,12].

Kind regards,

Christopher Strablegg

Reviewer 2 Report

Title: 

-       The title reads rather long and bulky and must be improved from the language point of view. 

Abstract: 

-       While the content reads fine, the abstract should be revised from the language. 

Keywords: 

-       The last keyword is too general. Authors are advised to replace it.

Introduction:

-       While the introduction provides some information about AE, the introduction into artificial intelligence and machine learning falls rather short. Recently published review articles and original research work by Marian et al. are ignored. In this regard, you may refer to: 

-       -- Current trends and applications of machine learning in tribology—A review

-       -- The use of artificial intelligence in tribology—A perspective

-       -- Predicting EHL film thickness parameters by machine learning approaches

-       The novelty of the presented study should be better worked out. 

-       The last paragraph of the introduction should be streamlined and improved.

Experimental method: 

-       More experimental information regarding the substrates, oils and testing conditions must be included. The following statement remains unclear “Friction minimizing polymer coated”

-       What means oil 1-16?

-       What are the chemical compositions of all substrates?

-       What were the surface roughness of the substrates? Hardness?

-       Figure 4 and Figure 5 should be combined. 

-       Figure 6 looks rather like a flowchar or poster. Please make a scientific figure out of it. 

-       Section 2.5 “2.5. Classifiers and machine learning tools” must be extended to provide all “experimental” details. 

-        

-       Figure 2 should contain a scale bar. 

Results and discussion: 

-       Figure 7 and Figure 8 should be combined. 

-       Please provide the captions of all figures. 

-       The following statement “Interestingly higher load levels in the test environment lead to lower RMS/friction power values.” Holds true only for a part of Figure 10. Please revise. 

-       The design and layout of Figure 11 should be rethought and revised. 

-       The manuscript contains a large number of figures, which interrupts the flow and readability. 

-       In contrast, almost no scientific discussion is presented. Referencing throughout the entire article is rather poor. 

-       Discussion should be extended. 

-       Conclusions should be separately presented.

Author Response

Dear Reviewer, thank you for your valuable input. Beneath I want to go into detail what was changed in the work to address your comments:

  • Title:
    The title reads rather long and bulky and must be improved from the language point of view.
    The title was shortened for better readability.

  • Abstract:
    While the content reads fine, the abstract should be revised from the language.
    Language and the text was revised to feature more conclusions and results, sentences were shortened to boost readability.

  • Introduction:
    While the introduction provides some information about AE, the introduction into artificial intelligence and machine learning falls rather short. Recently published review articles and original research work by Marian et al. are ignored.
    The literature stated from you provides a noteworthy insight into the topic of machine learning in tribology and gives a good example of the capabilities of the technique. All 3 papers were included in the introduction in references [13,14] (Review, Perspective)) and [15] (EHL parameters) in Lines 52-57.

    The novelty of the presented study should be better worked out.
    From Lines 62-73 the novelty of this work was better pointed out then in the previous manuscript, also in comparison to already existing work.
    “In this work the task is to predict the friction power of a Ring-on-Disc (RoD) test sys-tem with acoustic data in combination with a machine learning based method. Compara-ble work was done already predicting COF [23, 24] or modelling elasto-hydrodynamic lu-brication friction by machine learning [25], but always with a limited scope in the variation of materials, lubricants or test conditions. The challenge in this work was to get a fine differentiation between the possible prediction results over a wide range of sliding materials, oils & tests conditions. The novelty is to combine spectral and event-based data from non-continuous high-speed AE measurements, feeding the classifier, with low-speed, but continuously sampled AE data in form of the RMS value, to get a step-free and continuous prediction for the friction power.”

    The last paragraph of the introduction should be streamlined and improved.
    The last paragraph from Lines 73-79 was improved language wise and streamlined.

  • Experimental method: 
    More experimental information regarding the substrates, oils and testing conditions must be included. The following statement remains unclear “Friction minimizing polymer coated”
    Further information to exact chemical composition of sliding materials and oils was not the scope of this work, to goal was simply to provide a wide range of combinations. The “Friction minimizing polymer coated” now reads Polymer coated bearing material”. In the caption of figure 4 and in the text in Lines 161-171 additional information on how the test continuous after the cut of point in Figure 4 at 130 min is added: while the whole test would last until 50 MPa maximum load, with load increments of 1 MPa for the shock-loads, like in the first 120 min.” (Lines 165-166). More information on how a typical test performs and what the other measured values are interpreted can be found now in a revised section in Lines 259-273.

    What means oil 1-16?
    17 different oils were used in this study, the information in brackets after every material description show how many and what oils are used. For example the AlSi samples were only tested with one oil (oil 16), which was also used for the Al/AlSn20/AlSn6 samples (were oil 16 is also stated).

    What are the chemical compositions of all substrates?
    See two paragraphs above.

    What were the surface roughness of the substrates? Hardness?
    Surface roughness values are now mentioned in line 152-155.

     Figure 4 and Figure 5 should be combined.
    Combined, now it is only Figure 4.

    Figure 6 looks rather like a flowchart or poster. Please make a scientific figure out of it.
    The stated figure was changed a bit, adding axis labels to the diagrams. Other than that, I would like to keep it that way, because it should be essentially an easy to read work-flow chart of the data preparation. In the caption it new reads also as “Work-flow chart”.

     Section 2.5 “2.5. Classifiers and machine learning tools” must be extended to provide all “experimental” details. 
    In Lines 241-242 it was noted that no surrogate decision splits were allowed, otherwise no other manual hyperparameter optimization was done.
    In Lines 243-245, I now attached a note that the kernel scale is optimized automatically by the program (no additional user input was needed for that). The penalty factor is influenced by the box constraint level (added to the text).
    For the KNN in Lines 246-247 it is explicitly written now that no further manual hyperparameter optimization was done.

    Figure 2 should contain a scale bar.
    The picture of the samples was done at an angle, so a scale bar wouldn’t give good information. But, the size information of the samples was added in the caption of the Figure.

  • Results and discussion: 
    Figure 7 and Figure 8 should be combined.
    Combined, now they are Figure 6.

     Please provide the captions of all figures.
    All Figures have captions. Otherwise, I may not have understood your comment, right?
    Or was meant to provide a list of the figures and captions at the end of the work?

    The following statement “Interestingly higher load levels in the test environment lead to lower RMS/friction power values.” Holds true only for a part of Figure 10. Please revise. 
    The whole paragraph from Lines 286-292 was revised and more description was added to clarify the statement.
    “Interestingly, the increase in load levels in both test environments lead to lower RMS/friction power values. For example, in Figure 7 (b) at 135 min test time the load is in-creased (visible in Figure 6 (b)) and the AE RMS/friction power value decreases. Likewise, in the shock-load test of Figure 7 (a) the short regions with higher load feature lower AE RMS/friction power values. So, in theory the tribological processes have changed in a way that they are emitting less intense AE. This change could also be seen in the spectral com-position of the signal (see Figure 8).”

    The design and layout of Figure 11 should be rethought and revised. 
    The figure was separated into a new figure (Figure 9), which features better readability and a colour bar, and a table (Table 4) were all the TPR and FNR rates are listed.

    The manuscript contains a large number of figures, which interrupts the flow and readability.
    With the improvements already suggest by you the number of Figures is reduced and better readability should be given.

     In contrast, almost no scientific discussion is presented. Referencing throughout the entire article is rather poor. 
    Discussion and conclusion are now separated titles, both were expanded. From Line 450-465, the used frequency range is discussed more in depth. Also, the description of the impacts of the spectral data and the hit-rate on the classification is expanded there.
    In 475-484 it was better detailed how the classifiers helps one to deduce the impact of the different friction mechanism (from AE data). Information to known effects and their connection to certain frequency regions, was shortly mentioned now with the connected references of our previous work.
    In Line 485-508, more limitations and problems of the classification system are mentioned, as well as proposals for the future (for partly fixing them).
    In Line 511-528 the novelty of this work was depicted more directly and the impacts of the different data for the classifiers was also mentioned. Every type of classifier was again stated and the reliability of the system was stated, mentioning the two examples presented in Figure 10 & 13.

    Discussion should be extended. 
    See above.

    Conclusions should be separately presented.
    Done see above.

    Kind regards,
    Christopher Strablegg

Reviewer 3 Report

This study predicts the friction power of a given system (lubricated ring-on-disc geometry), independently of used material and oil, from the acoustic emissions emitted of the system alone, indirectly through a machine learning approach. Generally, the research ideas are interesting and may require further improvement in structure and some details. It’s suggested to consider the following aspects:

1. Please review and improve the English language and grammar.

2. Abstract: The abstract is expected to include a brief digest of the research, that is, new methods, results, concepts, and conclusions only. The abstract needs to be more focused and achievements needs mentioned clearly. At the moment abstract is more like an introduction than abstract. Please add some information from the conclusion (quantifications).

3. The object of the paper is the friction power of the system for acoustic emission data prediction. It is suggested to add the current research work in this field in the introduction section and to condense and summarize the innovation of the proposed method.

4. Please check the paragraphing.

5. Please check the revision of the title number. "2.5.Classifiers and machine learning tools" should be titled 2.6.

6. How to set the hyperparameters of support vector machine, such as penalty factor and kernel parameter, please give specific parameters.

7. In Figure 7 and Figure 8, several curves appear, but only AE RMS and Fiction power are described in the text, what are the meanings of the other curves and please explain them.

8. Some of the bullet points on the conclusion are simplistic; Please try to emphasize your novelty, put some quantifications, and comment on the limitations.

Author Response

Dear Reviewer, thank you for your valuable input. Beneath I want to go into detail what was changed in the work to address your comments:

  1. Please review and improve the English language and grammar.
    A lot of typos and grammar was corrected.
  2. Abstract: The abstract is expected to include a brief digest of the research, that is, new methods, results, concepts, and conclusions only. The abstract needs to be more focused and achievements needs mentioned clearly. At the moment abstract is more like an introduction than abstract. Please add some information from the conclusion (quantifications).
    In line 18-21 the text in the abstract was changed to: “. The method allows for the quantifiable and step-free prediction of absolute values of friction power with an accuracy of 97.6 % on unseen data with a weighted K-nearest-neighbor classifier at any point in time of an experiment.  The system reacts well to rapid changes in friction conditions due to changes in load and temperature.”
    Also from line 14-17 the three prominence levels for the hit-rate and the used types of classifiers were noted.
  3. The object of the paper is the friction power of the system for acoustic emission data prediction. It is suggested to add the current research work in this field in the introduction section and to condense and summarize the innovation of the proposed method.
    All in all, 6 new references were included in the introduction, [13,14] & [15] concerning machine learning in tribology and [23-15] concerning prediction of COF & simulating an EHD systems friction.
  4. Please check the paragraphing.
    Yes, there were some mistakes in the paragraphing, hopefully there are fixed now.
  5. Please check the revision of the title number. "2.5.Classifiers and machine learning tools" should be titled 2.6.
    Fixed, now it is 2.6.
  6. How to set the hyperparameters of support vector machine, such as penalty factor and kernel parameter, please give specific parameters.
    In line 243-245, it was already noted that the kernel was a cubic function, also I now attached a note that the kernel scale is optimized automatically by the program (no additional user input was needed for that). The penalty factor is influenced by the box constraint level (added to the text), which a set to the default value of 1.0, which was stated. Also, the learning method is given.
  7. In Figure 7 and Figure 8, several curves appear, but only AE RMS and Fiction power are described in the text, what are the meanings of the other curves and please explain them.
    Due to other changes in the document the graphics are now combined and numbered Figure 6. Lines 260-275 now better reflect all the values in the graphs:
    “Data from both tests was used to generate the training data for the classifiers. When looking at the friction power (cyan) and the AE RMS value (magenta) at both tests one can clearly see the connection between the two values. Of course, also the COF (black) follows a similar trend. The measured normal load (green) shows the intendent steps already shown in Figure 4. The temperatures (dotted red = specimen temperature, solid red = oil bath temperature) react as expected to the load increases in both tests, temperate gets higher, specimen temperature sees a stronger increase due to the friction heat. The contact potential (grey) measures the electric potential between ring and disc, 50 mV is the maxi-mum value and means no electrical contact, while zero means direct unshielded contact between friction partners. The load spikes in the test in Figure 6 (a) go hand in hand with a collapse of the contact potential. There the buildup of tribological layers from oil additives is reduced, which exposes the surfaces to more direct contact. In-situ wear (dark blue) is also recorded and measures the relative separation of lower and upper specimen adapters. It visualizes the wear of materials while the test, good visible in Figure 6 (a) with the quite soft sliding material compared to the very hard sliding material in Figure 6 (b).”
  8. Some of the bullet points on the conclusion are simplistic; Please try to emphasize your novelty, put some quantifications, and comment on the limitations.
    Discussion and conclusion are now separated titles, both were expanded. From Line 450-465, the used frequency range is discussed more in depth. Also, the description of the impacts of the spectral data and the hit-rate on the classification is expanded there.
    In 475-484 it was better detailed how the classifiers helps one to deduce the impact of the different friction mechanism (from AE data). Information to known effects and their connection to certain frequency regions, was shortly mentioned now with the connected references of our previous work.
    In Line 485-508, more limitations and problems of the classification system are mentioned, as well as proposals for the future (for partly fixing them).
    In Line 511-528 the novelty of this work was depicted more directly and the impacts of the different data for the classifiers was also mentioned. Every type of classifier was again stated and the reliability of the system was stated, mentioning the two examples presented in Figure 10 & 13.

Kind regards,

Christopher Strablegg

Round 2

Reviewer 2 Report

Thank you very much for revising the manuscript.

The quality of the article has certainly improved. 

Before the article can be recommended for acceptance, the following minor aspects should be taken into consideration. 

- Please add scale bars in Figure 2. 

- The structure and layout of the article must be improved. In the results and discussion section, partially there are just added figures over several pages... Authors are still advised to reduce the number of presented figures and group them smartly together to improve the flowability and readability of the entire manuscript. 

Author Response

Dear Reviewer,
the work was adapted in the following manner to address your commenst:

  • Please add scale bars in Figure 2. 
    Scale bar was added.
  • The structure and layout of the article must be improved. In the results and discussion section, partially there are just added figures over several pages... Authors are still advised to reduce the number of presented figures and group them smartly together to improve the flowability and readability of the entire manuscript. 
    Previous Figure 7 & 8 were combined into one graphic, now revised Figure 7.
    Figures 10-12 were combined, now revised Figure 9.
    Figures 13-15 were combined, now revised Figure 10.
    Table 1 & Figure 5 were grouped together.
    All the changed should improve the readability of the paper.

Kind, regards

Christopher Strablegg